# The Role of the 145 Residue in Photochemical Properties of the Biphotochromic Protein mSAASoti: Brightness versus Photoconversion

**DOI:** 10.3390/ijms232416058

**Published:** 2022-12-16

**Authors:** Alexandra V. Gavshina, Ilya D. Solovyev, Alexander P. Savitsky

**Affiliations:** A. N. Bach Institute of Biochemistry, Federal Research Center “Fundamentals of Biotechnology” of the Russian Academy of Sciences, Leninsky Ave. 33, bld. 2, 119071 Moscow, Russia

**Keywords:** biphotochromic fluorescent proteins, brightness, photoconversion green to red, photoswitching, photochemistry of fluorescent proteins

## Abstract

Photoswitchable fluorescent proteins (FPs) have become indispensable tools for studying life sciences. mSAASoti FP, a biphotochromic FP, is an important representative of this protein family. We created a series of mSAASoti mutants in order to obtain fast photoswitchable variants with high brightness. K145P mSAASoti has the highest molar extinction coefficient of all SAASoti mutants studied; C21N/K145P/M163A switches to the dark state 36 times faster than mSAASoti, but it lost its ability to undergo green-to-red photoconversion. Finally, the C21N/K145P/F177S and C21N/K145P/M163A/F177S variants demonstrated a high photoswitching rate between both green and red forms.

## 1. Introduction

SAASoti was isolated as a green-to-red photoconvertible protein [1]. Its monomeric form was successfully applied in the PALM (photoactivated localization microscopy) [2] and FCS (fluorescence correlation spectroscopy) [3] methods, and subsequently, it was found that its green form can be reversibly photoswitched to the dark state [4,5], which makes SAASoti a biphotochromic protein. Sequence alignment with other photoconvertible proteins (Figure 1) revealed that there stands a lysine residue in the 145th position (K145, SAASoti numbering), which is rather conservative for this protein family. Photoswitchable FPs are essential tools for conducting super-resolution techniques, e.g., PALM [6], RESOLFT [7] (reversible saturable optical fluorescence transitions) [8,9], SOFI [10], and the ability to simultaneously perform photoconversion and photoswitching increases the number of tasks that can be solved by using biphotochromic proteins. As an example, it is possible to observe the reallocation of labeled objects in cells with high spatial resolution [11]. The amino acid residues in the 163rd and 177th positions were often mutated in order to increase the photoswitching efficiency [12,13]; however, the resulting proteins sometimes had small molar extinction coefficients. In contrast to that, the replacement in the 145th position resulted in the increased brightness (and also molar extinction coefficients) in the case of the Dronpa family [13]. For this reason, it was decided to perform a homologous K145P substitution on the SAASoti protein and estimate its effect on the brightness of fast switchable M163X and F177X forms. We also incorporated C21N substitution, as it abolishes oligomerization at higher concentrations by preventing S-S bridge formation [14].

## 2. Results

Site-directed mutagenesis was used to generate K145P, C21N/K145P, C21N/K145P/M163A, C21N/K145P/F177S, and C21N/K145P/M163A/F177S variants. After protein isolation and purification, we characterized their physicochemical properties and kinetic parameters in the reactions of photoconversion and photoswitching.

K145P mSAASoti had the maximum value of the extinction coefficient (ε) for the green form; the bright lemon green color was observed directly after the centrifugation of *E. coli* cells expressing K145P mSAASoti. The C21N/K145P mSAASoti variant had minimal differences in physicochemical properties in comparison to K145P mSAASoti (Table 1).

Green-to-red photoconversion under 400 nm illumination was observed for all the mutants except for the C21N/K145P/M163A variant. As can be seen from absorbance spectra presented in Figure 2, K145P introduction did not alter photoconversion at all, as after 10 min of irradiation, a peak 590 nm characteristic of the red mSAASoti appeared, while the absorption of the green form (509 nm) was naturally decreasing. As for C21N/K145P/F177S and C21N/K145P/M163A/F177S mutants, we also observed an increase in absorption in the 450 nm region, which is most likely associated with the protonated red form. The suggestion was confirmed later by the increased pKa value of the red form, as seen in Table 1. Violet light illumination during 10 min of the C21N/K145P/M163A mSAASoti only led to the irreversible fading of the green form without any formation of the red mSAASoti species.

We also measured fluorescence lifetimes (Table 2) of both green and red forms for the new mutant forms. Green forms of mutants with substitutions in the vicinity of the chromophore (M163, F177) had a bi-exponential nature (Equation (1)) of the fluorescence decay, and the values of the average fluorescence lifetime (τ_mean_) became shorter compared to the wild type mSAASoti. The bi-exponential dependency of the fluorescence decays may be caused by the existence of various radiating conformers at the ns scale. Fluorescence lifetimes of the red forms became longer than the original red mSAASoti in the case of new mutants.
(1)$$I\left(t\right)=A_{1}\times \exp\left(-k_{1}\times t\right)+A_{2}\times \exp\left(-k_{2}\times t\right)+c$$

On-to-off photoswitching of the new mSAASoti mutants was studied by 470 nm illumination of the protein solutions in cuvette and simultaneous registration of the emission spectra with the use of homebuilt equipment described below, followed by the kinetic analysis. As can be compared from the kinetic curves in Figure 3, K145P substitution did not affect the rate or efficiency of the on-to-off photoswitching. C21N/K145P/M163A demonstrated the maximum photoswitching rate (green line) of the green form (Figure 3). The red forms of C21N/K145P/M163A/F177S and C21N/K145P/F177S mutants also became photoswitchable, whereas C21N/K145P could not switch between fluorescent on- and dark off-states as the wild-type protein (Figure 3).

The fluorescence decreased under the 470 nm (or 550 nm) irradiation, obeying a bi-exponential law (Equation (1)) for all the mutants, except those with M163A substitution (Table 3). Interestingly, the first component is responsible for the switching, while the second one describes some fluorescence increase, as *A_2_* had the opposite sign.

Mean rate constants calculated as *k*__mut_ = (*A*_1_ × *k*_1_ + *A*_2_ × *k*_2_)/(*A*_1_ + *A*_2_) were normalized to the corresponding value (*k*__mean_ = (*A*_1_ × *k*_1_+*A*_2_ × *k*_2_)/(*A*_1_ + *A*_2_)) of the original mSAASoti green form in order to find the fastest mutants. Thus, the last two columns in Table 1 showed that the rate constant of C21N/K145P/M163A was 36.1 times higher than that of the original mSAASoti, C21N/K145P/F177S switched to the dark state only 9.8 times faster, and the red C21N/K145P/M163A/F177S and C21N/K145P/F177S forms had comparable photoswitching rates with the green wild type mSAASoti form. The combination of M163A and F177S substitutions in the C21N/K145P mSAASoti protein did not lead to an even faster switchable variant, i.e., no additive effect was observed, and all physicochemical parameters demonstrated average values between C21N/K145P/M163A and C21N/K145P/F177S variants (Table 1). To obtain mSAASoti variants with a high photoswitching rate of both green and red forms, we conducted site-saturated mutagenesis at the 163rd and 177th positions in the C21N/K145P mSAASoti gene using degenerate primers. We analyzed the following mSAASoti clones: C21N/K145P/M163X and C21N/K145P/F177X (X states for any amino acid residue). The selection was made by fluorescence screening. The colonies were illuminated with 470 nm light, fast photoswitchable variants were sent for sequencing to determine the appropriate substitution, and they contained M163F/C/I/V/P or F177T/L/C/Q/G substitutions. As can be seen from Table 1, for colonies expressing C21N/K145P/M163X and C21N/K145P/F177X proteins, we did not find variants superior to the C21N/K145P/M163A mSAASoti.

On-to-off photoswitching is a reversible process, and fluorescence recovery from the dark state can proceed either by 400 nm illumination (within seconds), as chromophore is captured in a protonated form in its dark state, or by thermal relaxation. Thermal relaxation was studied as the absorbance recovery of previously switched-off proteins as described earlier [5]. Relaxation kinetics can be described by a mono-exponential model (Equation (2)):(2)$$A=A_{0}\times \left(1-\exp\left(-k\times t\right)\right)+c$$

The incorporation of K145P to the C21N mSAASoti gene did not influence the relaxation rate, but the addition of M163A or/and F177S substitutions ‘returned’ the constant values to that of the wild-type protein (See Table 3, k__rel_). The maximum absorbance recovery in the off-to-on switching was observed for the variants containing M163A mutations, which is probably connected to the fact that M163A mutant forms are more oxidation photostable.

## 3. Discussion

The addition of the K145P substitution led to an increase of the molar extinction coefficient (ε) for the green form, while the red form obtained a lower value. K145P mSAASoti was characterized by improved brightness when expressed in *E. coli* cells, which may be associated with an increase in the extinction coefficient and a pKa shift to the region of lower pH values (5.8) (Table 1). The K145P substitution itself might shift the equilibrium between the forms, and the fraction of the properly maturated chromophore increased.

The P145I substitution was incorporated to the Padron protein, a Dronpa derivative that switches from a stable off- to on-state (positive photoswitching) [15]. To obtain a more hydrophilic interface, the P145R (Padron*) variant was also generated. It is characterized by a lower tendency for oligomerization. According to structural studies of the Padron 0.9 (Padron derivative) [16], P145 residue does not affect the change in the switching nature; however, it determines differences in structural rearrangements during chromophore switching and spectral changes. After P145L substitution, there was no serious displacement of the amino acid residues of the off-form in comparison with the on-state (Padron0.9 compared to Dronpa [17]), but strong distortions of the conjugated chromophore system were observed, and the protein matrix remained almost unchanged. Thus, it is worth noting that the rigidity of the β-barrel is not so critical for photoswitching, which is also confirmed by molecular modeling [18]. At the same time, proteins with P145 have large extinction coefficients, and due to similar quantum yield values, they have greater brightness (among the Dronpa variants). It is also interesting to note that the mechanism of the double bond isomerization of the chromophore itself may depend on the packing of the crystal [19]. The presence of a cavity in the chromophore region allows for the local rearrangement of residues during cis-trans isomerization of the chromophore [20]. As it was mentioned earlier, differences in the behavior of proteins with P145 are associated with the mobility of the protein skeleton. In the case of Padron [16], the shift of S146 leads to the situation that H193 cannot move into the cavity that Dronpa has; thus, R66 occupies its place, and the chromophore has to isomerize in a limited space. These key a.a. residues are also present in mSAASoti; therefore, we suppose that, in the case of SAASoti, it might proceed according to the same scenario.

Amino acid substitutions in the chromophore vicinity increase its mobility; therefore, it is possible to register multiple conformations of the chromophore in the ground state. This can explain the broadening of the absorption spectrum (Figure 2, green forms). We also observed multicomponent kinetics of fluorescence attenuation of mutant forms when studying the lifetime in the excited state. A decrease in the lifetime may indicate a possible quenching of the conformers of the destabilized chromophore. The emission spectra changed less, and possibly, the excited state was closer to the wild-type protein. Similar changes in spectral properties were observed for homologous proteins and their F173 and M163 mutants [21]. The blue shift in absorption spectra was explained by an increase in the proportion of imidazole moiety in equilibrium with the phenolic form of the chromophore.

Interestingly, the M163A substitution to the C21N/K145P protein abolished the green-to-red photoconversion, whereas the C21N/K145P mSAASoti mutant had both green and red forms with relatively high ε values. In other words, this cumulative allosteric effect was the result of the triple substitution, and only the methionine residue was located in the vicinity of the chromophore. The K145P substitution in combination with M163A and F177S also dramatically affected the efficiency of the photoconversion, but it still occurred, as opposed to the C21N/K145P/M163A mutant. The C21N/K145P/M163A mutant demonstrated the maximum photoswitching rate of the green form without the ability of green-to-red photoconversion. The bi-exponential nature of photoswitching was observed earlier [5]. Here, we suggest that M163 might play a special role in the appearance of the protein fraction with different kinetic behavior in response to the radiation light. It should be noted that Skylan [22,23] and mGeos [24], photoswitchable variants of mEos2 that lack the ability of green-to-red photoconversion, were obtained by replacing H62, a very conservative residue in the chromophore forming triad for the photoconvertible FPs, whereas C21N/K145P/M163A mSAASoti with the histidine could not be photoconverted. To sum up, C21N/K145P/M163A can be utilized in super-resolution techniques such as fast photoactivation-localization microscopy [25], SOFI (super-resolution optical fluctuation imaging) [22,26], RESOLFT [27], and NL-SIM [23] as a green reversibly fast photoswitchable mSAASoti variant with more photoswitching cycles and better fluorescence recovery, since 400 nm light applied to the on-state regeneration would not induce the competing photoconversion reaction and photodestruction.

## 4. Materials and Methods

### 4.1. Construction of Mutants

Site-directed and site-saturated mutagenesis was carried out by the overlap PCR method [28], as described in [2]. The synthesis of primers was purchased from Evrogen, LLC (Moscow, Russia).

Recombinant proteins were expressed and purified as described in detail previously [14].

### 4.2. Colony Screening

The ligating mixture after overlapping PCR with degenerate primers was transformed into *E. coli* BL21(DE3) cells by electroporation and were seeded on agar plates, containing LB medium with 100 μg/mL ampicillin. Bacteria expressing different mSAASoti mutants were grown at 20 °C on Petri dishes, and colonies expressing fluorescent proteins were observed on the next day. The homebuilt installation (described below) was used to perform fluorescent screening. The colonies were analyzed by spectral and kinetic switching properties. After screening, sequencing of the most promising colonies (with maximum photoswitching rate) was performed by Evrogen, LLC.

### 4.3. Spectral and Phototransformation Properties Characterization Setup

The selection of promising clones expressed in *E. coli* colonies after site-saturated mutagenesis was carried out on an epifluorescence installation with a spectrometer.

Olympus BX-43 body was used as a base for the device. Four Thorlabs LEDs (390, 450, 470, and 560 nm) were collimated by the achromatic condenser lenses Thorlabs ACL2520-A and coupled by the three dichroic mirrors Thorlabs DMLP425R, DMLP490R and Edmund Optics #67-078, 458 nm long pass. Spectral bands were carried out by the bandpass filters Thorlabs MF390/18 and Chroma ET448/19x, ET470/24m, ZET561/10x, or ET560/25x. We used the Köhler scheme to obtain a more homogenous light beam after the microscope objective. The light beam after collimating lenses was focused on the objective back plane by the achromat lens (Thorlabs AC254-125-A). Then, light after the achromat was reflected by the 50/50 beam splitter on the objective. The fluorescence image was projected on the CCD camera after the beam splitter through a tube lens after Chroma 500LP and ZET562NF Notch filter. The camera was used to focus and orient the samples. At the same time, an achromat lens focused the image on the entrance slit of the Avesta ASP-75 Spectrometer through the second 70/30 beam splitter. LEDs were controlled by the Thorlabs LEDD1B driver and homemade USB DAC with self-written Python software. It allows for the switching of LEDs with 1 ms time resolution. We obtained 282.4, 528.4, 706.1, and 45.6 (ZET561/10x) mW/cm^2^ maximum light power densities for the 390, 450, 470, and 560 nm, respectively, after the 20x/0.4 NA Olympus PlanApo objective.

Absorbance and fluorescence spectra were registered as described in [2], and pKa and molar extinction coefficient values were calculated according to equations presented in the same work [2].

The PicoQuant Fluotime 200 spectrometer with a TCSPC (time-correlated single-photon counting) system was used to obtain FPs fluorescence lifetimes. The green form was excited by a 480 nm pulsed laser and the red form by 532 nm. Emission for both forms was registered at the maximum emission wavelength. Acquired data were analyzed using PicoQuant FluoFit software.

### 4.4. Phototransformation Kinetics

Isolated and purified mSAASoti variants (C21N, M163A/T, F177S, C21N/M163A, C21N/F177S, C21N//M163A/F177S) were irradiated according to the scheme presented in [5] when studying photoconversion and photoswitching.

Colonies (indicated in Table 1 with *), as well as a reference, were irradiated with 470 nm light with a power density of 437 mW/cm^2^, switching to the on state and photoconversion with 400 nm 148 mW/cm^2^ light for colonies. We used colonies with mSAASoti variants as reference kinetic data. Kinetic data analysis was carried out, as previously reported [14].

Thermal relaxation was studied by recording absorption spectra in the time of preliminarily switched-off, with blue light protein solutions using a Cary 60 spectrophotometer (Agilent). Kinetic curves were plotted based on the values of the absorption maxima of the anionic form. Data analysis was performed with Origin 8.5 software package.

## Figures and Tables

**Figure 1 ijms-23-16058-f001:**
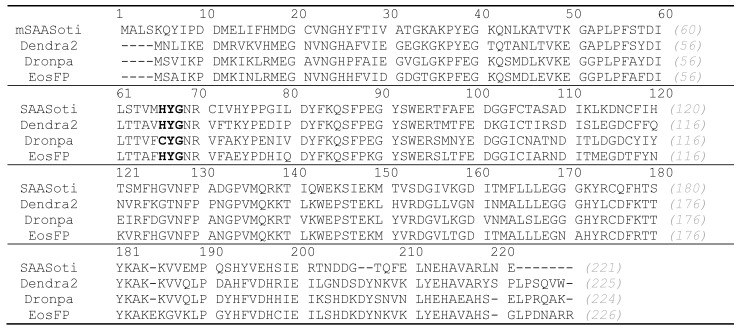
Multiple sequence alignment of mSAASoti with photoconvertible FPs. The numbering of all amino acid residues is given according to the SAASoti sequence.

**Figure 2 ijms-23-16058-f002:**
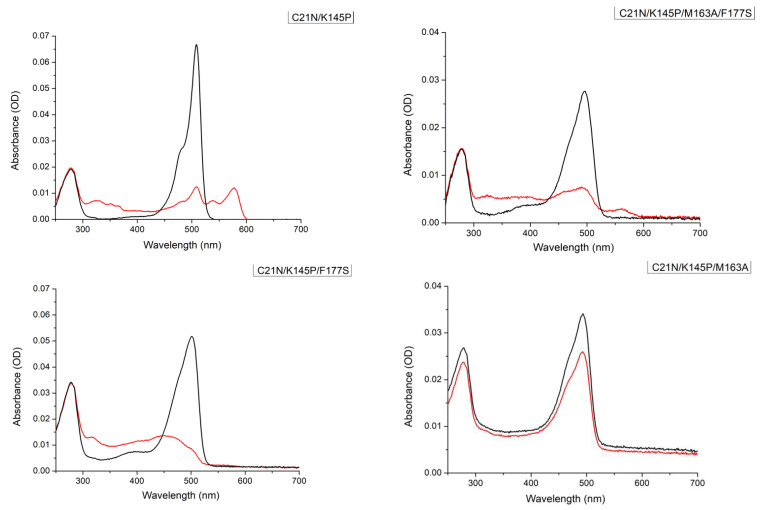
Absorbance spectra before (black line) and after (red line) 10 min of 400 nm illumination of the protein solutions measured in 20 mM Tris–HCl, 150 mM NaCl pH 7.4 for C21N/K145P, C21N/K145P/M163A/F177S, C21N/K145P/F177S, and C21N/K145P/M163A mSAASoti variants.

**Figure 3 ijms-23-16058-f003:**
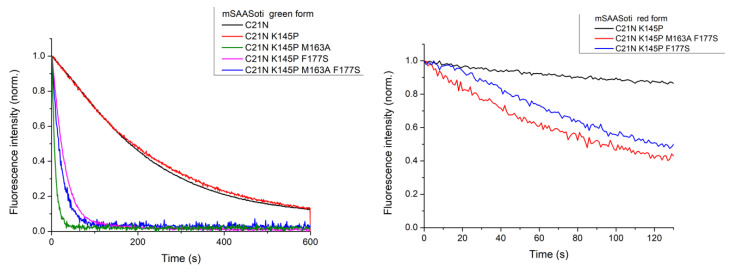
Photoswitching kinetics of the green (**left**) and red (**right**) mSAASoti mutant forms measured during 470 nm and 550 nm illumination, respectively.

**Table 1 ijms-23-16058-t001:** Fluorescent characteristics of mSAASoti and its mutant forms.

Mutant Form	Green Form λex, nm	Red Form λem, nm	ε G/R /1000 M^−1^ ×cm^−1^	pKa G/R	Green *k__mut_*/ *k__mean_*	Red *k__mut_*/ *k__mean_*
**mSAASoti [2]**	509/519	578/589	75/24	6.3/6.6	1.0	–
**C21N [14]**	509/519	579/590	82/25.4	6.4/7.5	1.0	–
**K145P**	509/519	578/589	88/20.5	5.7/6.7	1.0	–
**C21N/K145P**	509/519	578/589	87/16	5.8/n.a.	1.1	–
**C21N/K145P/M163A**	494/515	–/–	43/–	6.2/–	36.1	–
**C21N/K145P/F177S**	501/518	568/584	47/0.3	6.3/n.a.	9.8	1.1
**C21N/K145P/M163A/** **F177S**	496/516	560/585	46/3.2	6.0/n.a.	13.4	1.2
**C21N/K145P/M163F**					0.73 *	
**C21N/K145P/M163C**					13.6 *	
**C21N/K145P/M163I**					15.5 *	
**C21N/K145P/M163V**					13.6 *	
**C21N/K145P/M163P**					1.1 *	
**C21N/K145P/F177T**					4.3 *	
**C21N/K145P/F177L**					1.0 *	
**C21N/K145P/F177C**					4.7 *	
**C21N/K145P/F177Q**					3.2 *	
**C21N/K145P/F177G**					12.8 *	

– no photoswitching (or photoconversion) observed; n.a.–measurements were not carried out; the (*) symbol notes the data obtained for colonies. The protein solutions were prepared at pH 7.4 (20 mM Tris-HCl, 150 mM NaCl).

**Table 2 ijms-23-16058-t002:** Fluorescence lifetimes of mSAASoti and the mutants calculated according to Equation (1). Excitation wavelengths were 480 nm and 532 nm for green and red forms, respectively. Emission for both forms was registered at the maximum emission wavelength. The protein solutions were prepared at pH 7.4 (20 mM Tris-HCl, 150 mM NaCl).

	Green Form					Red Form
	** *A* _1_ **	**τ, ns**	** *A* _2_ **	**τ, ns**	**τ_mean_, ns**	**τ, ns**
**mSAASoti**		3.3			3.3	3.8
**C21N/K145P**		3.4			3.4	4.3
**C21N/K145P/M163A/F177S**	1600	3.05	3800	1.55	1.99	4.2
**C21N/K145P/F177S**	7000	3.28	1300	1.75	3.04	4.0
**C21N/K145P/M163A**	4200	2.9	1900	1.6	2.50	

**Table 3 ijms-23-16058-t003:** Kinetic parameters in the on-to-off photoswitching reaction of the green form measured in 20 mM Tris-HCl, 150 mM NaCl, pH 7.4 buffer, calculated according to Equation (1) for different SAASoti mutant forms; recovery (%) was calculated as the ratio of the absorption after the relaxation and before the switching; k_rel_ corresponds to the reaction of thermal relaxation (off-to-on switching).

Green Form	*A* _1_	*k* _1_	*A* _2_	*k* _2_	Recovery, %	*k__rel_*, min^−1^
**mSAASoti [14]**	1.25	0.0059	−0.34	0.0135		0.022
**C21N [14]**	1.19	0.0056	−0.28	0.016		0.01
**C21N/K145P**	1.01	0.0043	−0.06	0.03	76	0.01
**C21N/K145P/M163A**		0.13			100	0.022
**C21N/K145P/F177S**	1.12	0.038	−0.11	0.15	84	0.024
**C21N/K145P/M163A/F177S**		0.048			97	0.021
**Red form**	* **A** * ** _1_ **	* **k** * ** _1_ **	* **A** * ** _2_ **	* **k** * ** _2_ **		
**C21N/K145P/F177S**	275	0.010	−46	0.09		
**C21N/K145P/M163A/F177S**		0.011				

## Data Availability

The data presented in this study are available on request from the corresponding author.

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
