# Peer review of "The Role of the 145 Residue in Photochemical Properties of the Biphotochromic Protein mSAASoti: Brightness versus Photoconversion"

_ijms, 2022, doi:10.3390/ijms232416058_

Round 1

Reviewer 1 Report

The manuscript by Gavshina and co-authors provides a description of photophysical and photochemical properties of a series of new photoswitchable mutants of biphotochromic protein mSAASoti. an additional advantage of these fluorescent proteins is that in addition to bright-to-dark photoswitching,  some of them are able to photoconvert from green to red form which makes it possible to use them in a larger number of tasks in super-resolution microscopy of cells.

I have a few suggestions for improving the manuscript.

1. Numbering of amino acid sequences in Table 1 should be corrected. It looks like all numbers ending by 0 (except for the first column) should be changed to the corresponding numbers ending by 1, except for the last column. Also, the numbers should be shifted such that the digit 1 will right above the position of the first amino acid in a column.

2. In the description of the fluorescence lifetime change upon substitutions (p. 3, line 73) it would be good to mention compared to what the fluorescence lifetime became shorter. Is it compared to mSAASoti? In the table 3, two different lifetimes should have indexes 1 and 2, similarly to the amplitudes. The authors should specify what lifetime becomes shorter, tau1, tau2, or average (and present the average in the Table).

3. Based on the structure, is there any explanation of why there are 2 lifetime components for certain mutants and only one in others?

4. At the end Discussion it is desirable to add a sentence or with possible answer to the question raised in the title of the paper. In particular, what is the role of 145 residue? Is it more important for brightness of photoconversion? Also, the meaning of the title is not clear and should be re-phrased. For example, one possibility could be "The Role of 145 Residue in Photochemical Properties of the Biphotochromic Protein mSAASoti: Brightness versus Photoconversion." 

Reviewer 2 Report

I see no scientific or technical difficulties associated with this paper.  The results would be of interested to a limited spectrum of fluorescent protein chemists. The manuscript does have to be rewritten by a native English speaker.

In line 42  I don't see "colored yellow" residues in the Figure.

Here I have rewritten the abstract

Abstract:Photoswitchable fluorescent proteins (FPs) have become indispensable tools for studying life sciences. mSAASoti FP, a biophotochromic FP is an important representative of this protein  family. We have created a series of mSAASoti mutants in order to obtain fast photoswitchable  variants with high brightness. K145P mSAASoti has the highest  molar extinction coefficient of  all  SAASoti mutants studied, C21N/K145P/M163A switches to the dark state 36 times faster than mSAASoti, but it  lost its ability to undergo green-to-red photoconversion, finally the C21N/K145P/F177S and  C21N/K145P/M163A/F177S variants demonstrate a high photoswitching rate between both green and red forms.

Reviewer 3 Report

Following sequences of other relative photoconvertible (pc) and photoswitchable (ps) proteins, the authors constructed several mutants of biphotochromic fluorescent protein (FP) mSAASoti and analyzed their photochemical properties. Mutation of lysine 145 to proline present in other FPs of the family increased the molecular brightness of mSAASoti, while the brightness of its red form got unfavourably lower. Two fast green-to-dark ps mutants of mSAASoti were shown with additional fast check for other mutants in respective residues.

Though such analyses is certainly necessary for development of more suitable pc/ps-FPs derived from mSAASoti, the outcome of the study is bit incremental and would have much higher impact if mutagenesis attempts will be broader/more systematic. Though use of generated fast photoswitching mutants for PALM, SOFI, etc. was suggested, from the discussion is still unclear in what are current mSAASoti mutants superior to other well established pc/ps-FPs. Finally, rather inconclusive answer is provided regarding the main question posed in tittle of the manuscript?

Technically, the rationale and results of performed experiment could be better described (especially the last part) and all tables a should be better explained in their footnotes.

Minor points:

2 Likely clearer: The role of Residue 145 in...

194 begginning of the sentence unclear
196 mcg => mg
197 Petri dishes do not grow:-)

247 Delete &amp

Table 1. No sequence in yellow highlighted, last row of sequences shifted against aa numbers!

Table 4. Why (14) 
